# A Study on the Derivation of Atmospheric Water Vapor Based on Dual Frequency Radio Signals and Intersatellite Communication Networks



**Ramson Munyaradzi Nyamukondiwa \*, Necmi Cihan Orger, Daisuke Nakayama and Mengu Cho**

Laboratory of Lean Satellite Enterprises and In-Orbit Experiments (LaSEINE), Department of Electrical and Space Systems Engineering, Kyushu Institute of Technology, Kitakyushu 804-8550, Japan; orger.necmi-cihan397@mail.kyutech.jp (N.C.O.); nakayama.daisuke199@mail.kyutech.jp (D.N.); cho.mengu801@mail.kyutech.jp (M.C.)

\* Correspondence: munyaradzi.nyamukondiwa-ramson769@mail.kyutech.jp

**Abstract:** The atmospheric total water vapor content (*TWVC*) affects climate change, weather patterns, and radio signal propagation. Recent techniques such as global navigation satellite systems (GNSS) are used to measure *TWVC* but with either compromised accuracy, temporal resolution, or spatial coverage. This study demonstrates the feasibility of predicting, mapping, and measuring *TWVC* using spread spectrum (SS) radio signals and software-defined radio (SDR) technology on low Earth-orbiting (LEO) satellites. An intersatellite link (ISL) communication network from a constellation of small satellites is proposed to achieve three-dimensional (3D) mapping of *TWVC*. However, the calculation of *TWVC* from satellites in LEO contains contribution from the ionospheric total electron content (*TEC*). The *TWVC* and *TEC* contribution are determined based on the signal propagation time delay and the satellites' positions in orbit. Since *TEC* is frequency dependent unlike *TWVC*, frequency reconfiguration algorithms have been implemented to distinguish *TWVC*. The novel aspects of this research are the implementation of time stamps to deduce time delay, the unique derivation of *TWVC* from a constellation setup, the use of algorithms to remotely tune frequencies in real time, and ISL demonstration using SDRs. This mission could contribute to atmospheric science, and the measurements could be incorporated into the global atmospheric databases for climate and weather prediction models.

**Keywords:** total water vapor content; total electron content; frequency reconfiguration; intersatellite link; software-defined radio; radio sensing

## 1. Introduction

The atmospheric total water vapor content (*TWVC*) is a critical meteorological measurand for the hydrological cycle, weather forecasting, climate modeling, Earth's energy budget, and ozone chemistry [1]. Recently, the unprecedented global climate change crisis related to *TWVC* has been evidenced by frequent and intense disruptive weather patterns such as floods, droughts, extreme heatwaves, rising sea levels, massive snowfalls, storms, and stronger hurricanes [2,3]. The severity and impacts of these disasters are predicted to continuously jeopardize people, infrastructure, global economies, and the Earth's natural ecosystems. In the next decade, the devastating effects of these calamities are predicted to plunge millions of people into poverty, unravel hard-won development gains, and displace people from their homes [4]. These problems demand the invention of accurate, reliable, resilient, and adaptative remote sensing techniques that predict, monitor, and minimize the influence of *TWVC* with sufficient spatial coverage and temporal resolution.

### 1.1. Conventional Techniques Analysis

Conventional ground, air, and space technologies are being used to predict *TWVC* distribution. Terrestrial remote sensing techniques that use ground or upward-looking

observation infrastructure such as surface meteorology, Raman light detection and ranging instruments (R Lidar), differential absorption Lidar, microwave radiometry, and sun photometers have proven successful in mapping the *TWVC* distribution with high temporal resolution and adequate precision [1,5–11]. However, these techniques are usually immovable, costly, require continuous maintenance, and have small spatial coverage. Also, surface meteorology-related data are closely tied to land–air data and cannot sufficiently map the complete boundary of atmospheric water vapor. Moreover, meteorological sensors for microwaves, Lidar instruments, and sun photometers are greatly affected by rain, clouds, and the intensity of sun, respectively. These effects greatly limit the efficiency and effectiveness of using these techniques in measuring parameters of the atmosphere. Very-long baseline interferometry is another ground-based technique that can detect atmospheric water vapor based on the delay of a radio signal when observing extragalactic radio emitters such as quasars using at least two ground-based radio antennas [10,12]. Nevertheless, it is usually characterized by low temporal resolution depending on the operation schedule.

Meteorological station data from 1966 to 2017 and ERA5 reanalysis data from 1979 to 2020 over Russia were used to study the scaling correlation between daily precipitation extremes and surface air temperature (SAT) as represented by the Clausius–Clapeyron (C-C) equation [13]. Northern Eurasia has not been adequately examined, despite major temperature shifts and a rapid transition from large-scale to convective precipitation. Reanalysis data might have inconsistences and station data might have limited spatial coverage, especially in remote regions, potentially leading to spatial bias. Gaining insight into the mechanisms driving intense precipitation allows for improved weather predictions and climate projections that rely on enhanced satellite technologies for improved accuracy.

Polar orbiting satellites, radiosondes, interferometric synthetic aperture radar (InSAR), imaging spectroradiometers such as the moderate and medium resolution imaging (MODIS onboard Terra and Aqua and MERIS onboard ENVISAT), remotely piloted vehicles, instrumented aircrafts, and global navigation satellite systems (GNSS) satellites are among the air and space-inclined techniques utilized to detect and measure *TWVC* [1,14–22]. These methods proved to be convenient and have large spatial resolutions; however, the space measurement technologies have low temporal resolution and compromised accuracy compared to ground-based methods. Furthermore, the use of imaging spectroradiometers (MERIS and MODIS) is affected by clouds in addition to their low temporal resolutions [22].

Two distinct Iranian regions with diverse climate conditions were explored to investigate the impact of shifts in meteorological indicators such as water vapor and temperature on groundwater reserves [23]. The application of Sentinel-1A acquisitions, the InSAR technique, and advanced integration methods were implemented to determine meteorological indicators. Synoptic observations, meteorological data, the general circulation model, and the statistical downscaling model (SDSM) were all used to project the indicators. The prediction of groundwater levels was accomplished through a combination of groundwater level observations, water vapor data derived from GPS estimations, and an evapotranspiration index relying on an artificial neural network (ANN) [23]. However, the ANN model's accuracy and reliability are strongly dependent on the quality and amount of accessible data. Also, inadequate or inconsistent geodetic measurements and groundwater level data may further affect the analysis's robustness.

Optimal approaches that enhance the downscaling process of the function-based tomography technique sourced from GNSS satellites were suggested [24]. The dependency on Global Navigation Satellite System (GNSS) data and the related hardware infrastructure is one possible downside of adopting the function-based troposphere tomography approach for effective precipitation downscaling. This reliance on GNSS signals could pose challenges in areas prone to signal interference or regions with limited or disrupted satellite signal reception, such as densely forested regions and urban canyons. The precision and reliability of the tomography approach may be affected in such areas, resulting in inaccuracies in the downscaling process.

Hussein et al. [25] utilized copulas functions to depict the seasonal dependency of precipitation and temperature. Whilst they managed to capture the statistical dependency, they could not completely account for the complex physical mechanisms that drive these interactions in the atmosphere. Furthermore, copulas frequently assume stationarity, which implies that the interactions between variables remain constant across time. In reality, climate change and other variables may cause non-stationarity, which could impact risk prediction accuracy.

Geostationary (GEO) satellites such as the Geostationary operational environmental satellites (GOES) have been vital in effectively monitoring and tracking severe weather environments like storms and hurricanes in real time due to their high temporal resolution and ability to repeat observations over a specific area [26]. Nevertheless, for detecting water vapor, their high orbital attitudes contain large amounts of plasma. Additionally, the spatiotemporal interpolation methods used to derive water vapor from GEO satellites are computationally demanding, require sophisticated algorithms, and are likely biased by data anomalies [26]. Table 1 below summaries the accuracy and temporal and spatial resolutions of some of the air, space, and ground-based techniques below.

**Table 1.** Comparison of existing techniques that measure atmospheric water vapor.

| Technique | Observing Geometry | Temporal Resolution | Spatial Resolution | Accuracy | Conditions |
|---|---|---|---|---|---|
| Surface meteorology | ground | 1–60 min | few meters–tens of meters | few mm | affected by environment |
| Lidar | ground, air, and space | low–high depending on observing geometry | low–high | few mm | cloud-free sky |
| Microwave radiometers | ground, air, and space | 5–15 min | low–high | 1–5 mm | rain-free sky |
| Sun photometer | ground | few times with high solar illumination intensity | High | few mm | direct sunlight and clear sky |
| VLBI | ground | mins–days (depends on schedule) | very low (few sites) | few mm | none |
| Polar satellites | space | 6–12 h | 1–10 km | few mm–1 cm | none |
| Radio sondes | air | 1–4 times a day | low | 1–3 mm | none |
| Imaging spectroradiometers | space | MODIS (1–2 days) MERIS (3 days) | MODIS (250 m–few km), MERIS (300 m) | few mm–1 cm | cloud–free sky |
| Remotely piloted vehicles and Instrumented aircraft | air | depends on flight duration (few mins–few hrs.) | few meters–tens of meters | few mm | depends on weather |
| In-SAR | space | 6–12 days | high | 1–2 mm | none |
| GNSS satellites (radio occultation) | space | 1–60 min | high | 1–5 mm | none |
| GNSS satellites to standard GPS ground receivers | ground | 30 s–few mins | tens–hundreds of km | 1–5 mm | none |
| Geostationary satellites | space | Mins–hourly updates | one–tens of km | few mm–few cm | none |

Small satellites technologies such as CubeSats are also increasingly gaining relevance in climate and meteorology studies [27]. As a part of the space precision atomic-clock timing utility mission (SPATIUM), the Kyushu Institute of Technology (Kyutech) launched

and successfully operated two low Earth orbit (LEO) satellites, which are SPATIUM-I (2U CubeSat) and SPATIUM-II (1U payload on a 6U CubeSat), for a technology demonstration of ionospheric electron density mapping [19,28,29]. Even though the SPATIUM mission has an objective of three-dimensional (3D) ionospheric mapping for *TEC* using a constellation of small satellites, SPATIUM I and II missions demonstrated technologies related to a chip-scale atomic clock and onboard processing capabilities that can receive and a demodulate spread spectrum (SS) signal for time delay measurements between the ground station (GS) and satellite receiver. As a result, the two satellites have not yet investigated the influence of *TWVC* on the time delay in measurements, 3D mapping of the atmospheric parameters, and employing an orbital constellation in addition to intersatellite communication.

On a global scale, GNSS satellites have carried out modeling and mapping of the atmosphere's geophysical parameters, such as *TWVC*, *TEC*, temperature, and pressure [30]. The GNSS constellation utilizes either the carrier phase or the pseudo-range radio occultation techniques to estimate the signal delay as a result of the mentioned geophysical parameters. The measurements are deduced along the GNSS signal array paths [15,31,32]. Models such as numerical weather prediction (NWP) models, global ionospheric models (GIM) and global circulation models were developed to map and model the vertical and horizontal profiles of the atmospheric parameters from GNSS satellites [33–35]. However, the precision of *TWVC* and *TEC* from GNSS carrier waves is compromised because GNSS higher L-band frequencies are less affected by the atmospheric geophysical parameters. The GNSS satellites were not developed to monitor water vapor distribution or electron density but rather for locating or timing purposes. Therefore, their precision in determining *TWVC* or *TEC* is limited. Like GEO satellites, GNSS's high orbital altitudes have a large plasma component which influences measurement accuracy [19,36]. Furthermore, when estimating water vapor time delay from GNSS measurements, the satellites' antenna phase center (APC) is prone to fluctuations and they require correct APC modelling. Incorrect APC modeling could yield false and additional trends in *TWVC* profiles [37]. In addition, GNSS satellites determine atmospheric water vapor with a spatial resolution that varies within tens of kilometers based on the number of receiving stations and a temporal resolution of seconds to an hour [10]. Therefore, improving the accuracy and spatiotemporal resolution still remains an essential task for the measurements of *TWVC* in addition to the *TEC*.

### 1.2. Proposed Technique Analysis

Detecting atmospheric water vapor using low-frequency radio signals from a constellation of small LEO satellites is proposed as an alternative method. Radio signals through space are primarily influenced by atmospheric water vapor content and total electron density. Approximately 99% of *TWVC* is in the troposphere, whereas *TEC* plasma density is dominant in the ionosphere [38]. The troposphere is a layer in the atmosphere's neutrosphere region. *TWVC* and *TEC* profiles could be acquired using a satellite constellation with high spatiotemporal resolutions and better accuracy than the GNSS and other conventional technologies. As a result, small satellites could be considered due to their short development span, affordable costs, and effectiveness in completing space scientific missions. Low Earth orbit contains a lower percentage of plasma than the GNSS orbits. Unlike SPATIUM-I and SPATIUM-II missions, the influence of both *TWVC* and *TEC* are taken into consideration and *TWVC* is separated from *TEC*. Spread spectrum (SS) ranging signals which allow multiple signals to share the same frequency are transmitted from a network of satellites using intersatellite link (ISL) communication [19]. Software-defined radio (SDR) technology has been adopted to transmit and receive the SS signals. SDR technology is considerably effective, in that communication digital signal processing (DSP) hardware can be implemented as software; it supports a wide range of radio frequencies, and its architecture is reconfigurable, extensible, reprogrammable, and upgradable in real time [39]. The SS signals are modulated with binary phase shift keying (SS-BPSK) modulation because BPSK-modulated data travels long distances, and the original message is easily detected at the receiver due to the efficient properties of BPSK technology [40].

ISL technology has been employed for various satellite communication and formation flight applications, enabling ranging, commanding, timing control, and data transfer in a distributed network system [41]. In this proposal, the intersatellite ranging from satellites in different LEO orbital planes is performed in all directions to acquire three-dimensional mapping of atmospheric water vapor and ionospheric density along each line of transmission. Each satellite in the ISL network has a mission payload for *TWVC* derivation. The mission payload is designed to fit small satellites with limited power, mass, and size. To derive the water vapor and electron density in three-dimensional distributions, we solve the inverse problems [19]. For a given set of water vapor and electron density distributions, we integrate the two quantities along each measurement path and compare *TWVC* and *TEC* with the ones derived from the measurement. We then find the 3D water vapor and electron density distribution that best matches with the measurement. Figure 1 depicts the main notion of water vapor and electron density dispersion measured from a constellation of satellites. The *TWVC* and *TEC* are detected based on signal propagation time delay, and the delay of SS signals along the intersatellite ranging path is calculated onboard each satellite.

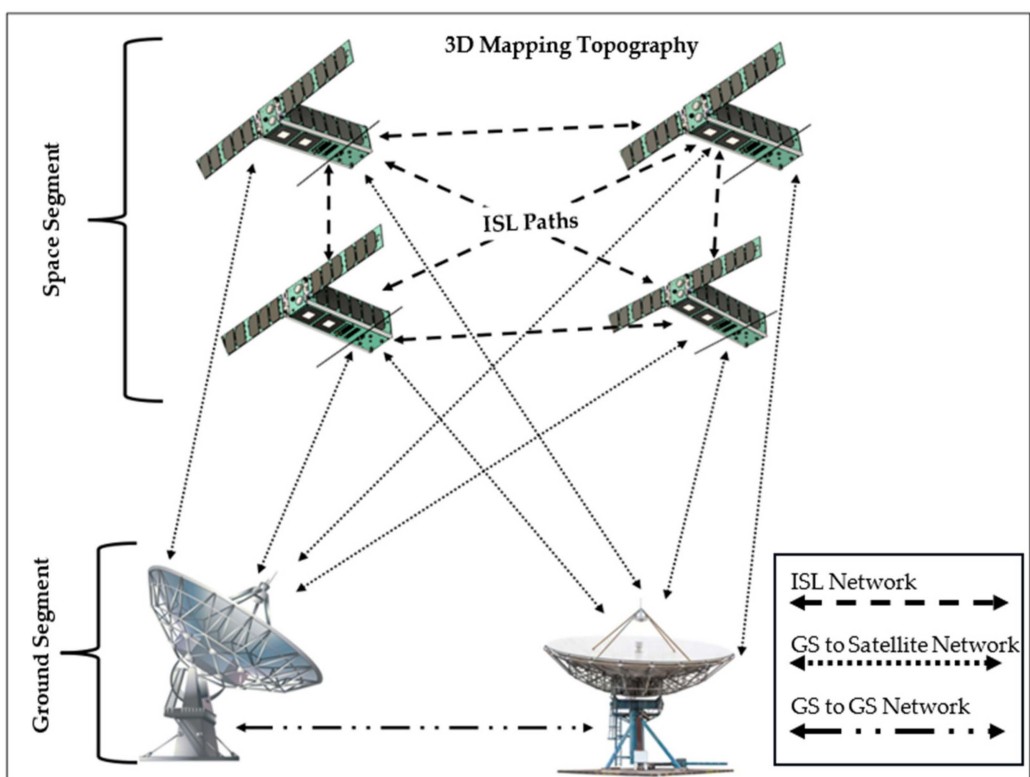

**Figure 1.** 3D mapping of atmospheric *TWVC* and electron density.

Since the total electron density varies inversely to the square of the radio signal frequencies, whereas *TWVC* does not, solving the inverse problem and manipulating multiple radio frequencies (RF) help to distinguish and separate *TWVC* from *TEC* [42]. Two remotely tuned algorithms that use extensible markup language remote procedure call (XML-RPC) and transport control protocol/internet protocol (TCP/IP) were coded to remotely reconfigure the SDR transmitter and receiver frequencies in real time [43,44]. The algorithms were built using GNU Radio digital processing blocks and the Python programming language. The algorithms achieve frequency translation within a second of reconfiguration. This is the shortest possible time to retrieve the mission data with high accuracy based on this method. Meteorological ultra-high frequency (UHF) bands were selected for communication since they are more vulnerable and influenced by the parameters of the atmosphere compared to GNSS L-band frequencies. The network of

ground stations (GS to GS network) uses GS to satellite networks to operate missions, control the satellites, and acquire mission data from satellites in the ISL network.

The following items summarize the primary goals of this study:

1. To deduce the *TWVC* measurement concept.
2. To design a satellite payload that measures atmospheric water vapor, determines the system requirements, selects the components and specifications, and conducts system interfaces and integration.
3. To implement a GNU radio-based SDR transceiver with both transmitting and receiving capabilities of SS ranging signals.
4. To perform ISL ranging. This is essential for 3D mapping when all LEO orbital planes will be considered.
5. To demonstrate dual frequency reconfiguration of SS ranging signals by remotely tuning SDR parameters during runtime onboard each satellite. This is required for mission measurement accuracy and to distinguish *TWVC* and *TEC*.
6. To eliminate instruments' clock offsets and errors as much as possible.
7. To simulate how the signal time delay due to water vapor and electron density can be estimated. This is vital in deducing the final *TWVC* measurement.

In terms of mission success criteria, achieving the above primary goals is a huge milestone which guarantees the viability of conducting a *TWVC* mission onboard the satellite based on this proposed method. Therefore, the purpose of this paper is to demonstrate the feasibility of the primary goals based on a ground test simulating an in-orbit demonstration mission. The ground test assumes satellites in the same orbital plane using two satellite payloads with the capability of both transmitting and receiving SS ranging signals. Therefore, this demonstrates the ISL network feasibility, and two frequencies are used for the frequency manipulation demonstration. The signals are time-stamped at both the transmitting and the receiving ends for time delay estimation. Unlike in the orbit scenario, for these ground tests there is no influence of atmospheric water vapor or *TEC*. For this reason, a simulation of signal time delay due to water vapor was implemented using a delay pulse generator. This functionality qualifies the ability of this system to determine the real water vapor content in the future. The most critical requirements of this study are listed below:

- The frequency reconfiguration time and data processing time should be $\leq 1$ s.
- A water vapor column of approximately a few mm and a delay measurement accuracy $\leq 100$ ns.
- The size of the constellation should be more than 1000 small satellites [19].
- The temporal resolution should be between 5 min and 15 min, whereas spatial coverage should be between 15 km and 4600 km.
- The payload should be able to fit within the limited constraints of power, size, and mass for a small satellite.

The novelty of the present paper is the use of SDR technology to deduce atmospheric water vapor delay based on radio signals, frequency manipulation, and the ISL network from a constellation of small satellites in LEO. Also, time stamps were implemented with RF switches in order to deduce the signal propagation time delay. The SDR capabilities were adopted to implement an automated software transceiver that performs digital signal processing (DSP) and frequency manipulation remotely in the shortest possible time. Instead of each satellite being composed of several radio devices that operate at different frequencies, only a single SDR transceiver is mounted onboard each satellite to perform the communication mission requirements and this reduces the cost and strain on the limited resources of small satellites. In this way, the size of individual satellites becomes smaller and the number of satellites in the constellation can be increased. In the present paper, we propose a constellation of small satellites to carry out 3D mapping of ionosphere and troposphere with improved spatiotemporal resolutions on the condition that the satellite constellation is large enough to cover a wider area. Moreover, the feasibility of ISL

ranging based on a low-cost commercial off-the-shelf (COTS) SDR transceiver and RPi microcomputer is uniquely demonstrated.

The ultimate goal of this study is to contribute to the advancements of climate and scientific studies of the atmosphere concerned with the influence of atmospheric water vapor and total electron content. In fact, prediction of *TWVC* assists in understanding the risks and impact of climate change and weather patterns so that mitigation and adaptation measures can be completed to safeguard life on Earth. The *TWVC* data acquired will be vital in the development of advanced climate and weather prediction models. Moreover, to a certain extent, this study finds its use in space communication and SDR technology applications.

In this paper, Section 2 details the theoretical concept of detecting both *TEC* and *TWVC* as well as the final derivation of atmospheric water vapor. Section 3 explains the design configuration for each satellite in the ISL network as well as procedures for determining time and frequency bands used for communication. Section 4 provides the setup utilized to demonstrate the ISL ranging, SS-BPSK software implementation, time delay detection, and elimination of SDR clock jittering errors. Section 5 illustrates the frequency manipulation algorithms and link budget analysis for the ISL network. Section 6 highlights discussions, and finally, the conclusion and suggestions for future work are given.

## 2. Theoretical Deduction of Atmospheric Water Vapor Content

In orbit, the satellites are in motion, so their positions and time change every second. Each satellite carries a GPS module which calculates position and time data. The position of a satellite relative to another satellite is known as the pseudo-range $(L_\rho)$. To detect the atmospheric water vapor, sounding is conducted near the Earth based on a constellation of satellites. Figure 2 shows how the atmospheric water vapor and total electron content can be deduced considering the communication of two satellites Sat *A* and Sat *B* in the ISL network.

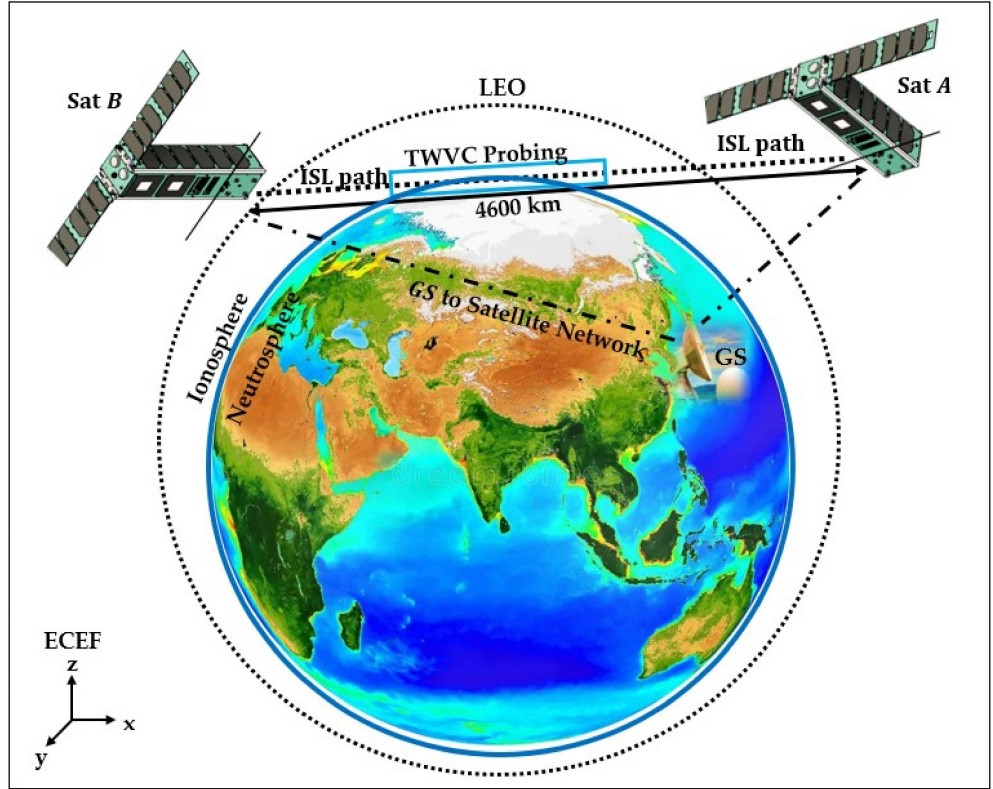

**Figure 2.** *TWVC* probing with the ISL network.

Knowing the satellites' location in the Earth-fixed coordinate system (ECEF) x, y, and z and the signal propagation time delay $\delta T$ (s), *TWVC* and *TEC* can be estimated. Assuming Sat *A* is in transmission mode (*Tx*) while Sat *B* is in reception mode (*Rx*), the combined signal propagation delay due to the troposphere's and ionosphere's influence at each position along the ISL path is obtained by comparing the transmitted and received signals. If the bit starting times or time stamp positions of the *Tx* and *Rx* signals can be detected, then the time delay can be calculated as described in Figure 3.

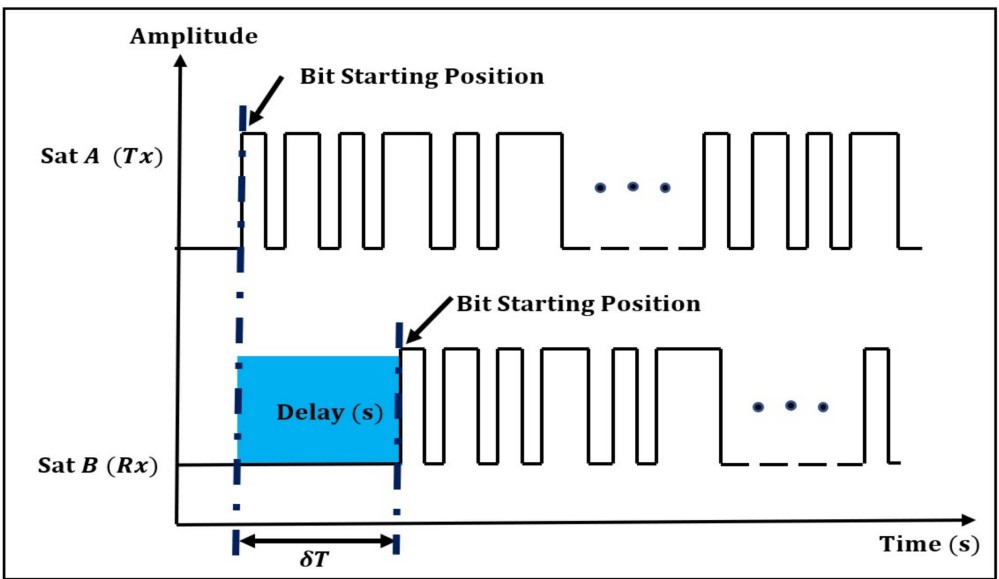

**Figure 3.** Time delay deductions.

In a more holistic approach, when the frequency of communication is $f_1$ (MHz), the time delay ($\delta T_1$) of the signal propagation is obtained based on all channel effects. These include the influence of integrated electron density and integrated total water vapor density along the signal ray between Sat *A* and Sat *B* at their positions in orbit denoted by *A* and *B* in the integrals, respectively. Mathematically, $\delta T_1$ is deduced as follows:

$$\delta T_1 = \frac{L_{\rho_1}}{c} + \frac{a}{f_1^2} \int_A^B \eta_{e_1}(l)dl + \frac{1}{\Pi} \int_A^B \eta_{TWVC_1}(l)dl + \Delta t_{Rx} - \Delta t_{Tx} + \in_{0_1} \tag{1}$$

where $L_{\rho_1}$ is the true range (m), $c = 3.0 \times 10^8$ ms$^{-1}$ is the speed of light, $\Delta t_{Rx}$ (s) is the Sat *B* clock receiver offset, $\Delta t_{Tx}$ (s) is the Sat *A* transmitter clock offset, and $\in_{0_1}$ (s) is the error in the range due to various other sources such as satellite instrument delays and relativistic clock corrections. The term $dl$ represents the path integral differential, $a = 40.3/c \approx 1.34 \times 10^{-7}$ m$^2$/s is a constant for *TEC* determination [45], and $\Pi$ [46–48] is an experimentally determined constant depending on air temperature. The term $\eta_{TWVC_1}$ (kg/m$^3$) is the water vapor density and its integral represents integrated total water vapor content along the ray path [47]. The total water vapor delay (s) primarily due to the tropospheric influence is deduced along the signal path based on integrated *TWVC* and $\Pi$. The term $\eta_{e_1}$ (electrons/m$^3$) is the total ionospheric electron density and its integral is the total electron content (*TEC*). The clock offsets are eliminated by using a precise clock as the clock source for both the transmitting and the receiving SDRs. The relativistic corrections and other known instrumented delays are compensated in the time delay detection software. Therefore, Equation (1) transforms to:

$$\delta T_1 = \frac{L_{\rho_1}}{c} + \frac{a}{f_1^2} \int_A^B \eta_{e_1}(l)dl + \frac{1}{\Pi} \int_A^B \eta_{TWVC_1}(l)dl \tag{2}$$

The true range ($L_\rho$) of each satellite relative to another satellite in the ISL network is computed from the altitude (m), latitude (°), and longitude (°) geodetic coordinates provided through the GPS module. When frequency is reconfigured to $f_2$ (MHz), time delay $\delta T_2$ is calculated as illustrated in Equation (3), where the satellite positions change from $A$ to $A'$ and from $B$ to $B'$, respectively.

$$\delta T_2 = \frac{L_{\rho_2}}{c} + \frac{a}{f_2^2} \int_{A'}^{B'} \eta_{e_2}(l)dl + \frac{1}{\Pi} \int_{A'}^{B'} \eta_{TWVC_2}(l)dl \tag{3}$$

It is important to note that frequency translation is not achieved instantly when the satellites are in motion. As a result, the water vapor content and electron density are determined based on signal propagation time delay between two positions ($A$ to $A'$) and ($B$ to $B'$) of each satellite in the ISL network using two different frequencies. Thus, to derive *TWVC* and *TEC*, the two time delays ($\delta T_1$ and $\delta T_2$) are compared as illustrated in Equation (4):

$$\delta T_1 - \delta T_2 = \frac{1}{C}\left(L_{\rho_1} - L_{\rho_2}\right) + a\left(\frac{1}{f_1^2}\int_A^B \eta_{e_1}(l)dl - \frac{1}{f_2^2}\int_{A'}^{B'} \eta_{e_2}(l)dl\right) + \\ \frac{1}{\Pi}\left(\int_A^B \eta_{TWVC_1}(l)dl - \int_{A'}^{B'} \eta_{TWVC_2}(l)dl\right) \tag{4}$$

In the closest range, when considering a constellation of thousands of satellites in different LEO orbital planes and directions, the difference between $L_{\rho_1}$ and $L_{\rho_2}$ is assumed to be at least 15 km. Assuming that the *TWVC* and *TEC* at the two positions of each satellite are the same, with *TEC* varying inversely to the square of the frequencies, this implies:

$$\int_A^B \eta_{TWVC_1}(l)dl \approx \int_{A'}^{B'} \eta_{TWVC_2}(l)dl = \int \eta_{TWVC}(l)dl, \text{ and} \tag{5}$$

$$\int_A^B \eta_{e_1}(l)dl \approx \int_{A'}^{B'} \eta_{e_2}(l)dl = \int \eta_e(l)dl \tag{6}$$

Substituting (5) and (6) into (4), *TEC* can be deduced as follows:

$$a\left(\frac{1}{f_1^2} - \frac{1}{f_2^2}\right)\int \eta_e(l)dl = (\delta T_1 - \delta T_2) - \frac{1}{C}\left(L_{\rho_1} - L_{\rho_2}\right) \tag{7}$$

$$TEC = \int \eta_e(l)dl = \frac{(\delta T_1 - \delta T_2) - \frac{1}{C}\left(L_{\rho_1} - L_{\rho_2}\right)}{a\left(\frac{1}{f_1^2} - \frac{1}{f_2^2}\right)} \tag{8}$$

However, the assumptions in (5) and (6) introduce an error in the measurement accuracy. This error is assumed to become negligible when the frequency transition can be achieved in 1 s or less. Also, by averaging measurements from multiple frequencies, the introduced error is anticipated to be very negligible. This is the other reason why multiple frequencies and frequency reconfiguration has been considered as a way of improving the mission measurement accuracy aside from distinguishing *TWVC* from *TEC*. When *TEC* variations become known, total water vapor content, *TWVC*, is computed from Equation (2) as follows:

$$TWVC = \int \eta_{TWVC}(l)dl = \Pi(\delta T_1 - \frac{L_{\rho_1}}{c} - \frac{a}{f_1^2}\int \eta_{e_1}(l)dl) \tag{9}$$

The key steps in this method are described as follows:

- Measuring the signal time delay by utilizing two frequencies between the satellites.
- Distinguishing the signal time delay due to *TEC* and *TWVC* by comparing the two frequency measurements.
- Collecting all the measurement data of *TEC* and *TWVC* within a time period.

- In the case of satellites in multiples orbital planes, obtain a 3D distribution of *TEC* and *TWVC*.
- Determining the most probable set of distribution that agrees with the measurement data of *TEC* and *TWVC*.
- Deducing only *TWVC* contribution by removing *TEC* values.

In line with Kyushu Institute of Technology satellite development projects, a constellation of more than 1000 satellites in low Earth orbit are expected to be launched for scientific studies of the atmosphere including for this mission to determine water vapor content in the atmosphere. These studies will be a continuation of the SPATIUM-I and SPATIUM-II missions which were demonstrated in orbit. The intersatellite ranging (ISL) communication network shall achieve a separation distance of 15 km to 4600 km and a temporal resolution of 5 to 15 min.

## 3. System Design Configuration

The system design architecture for each satellite in the ISL network is made using COTS components. The use of small standard satellites has been chosen for the satellite design because of their fast development time and low cost. The design configuration is divided into two parts, the satellite bus and the *TWVC* mission portions, as illustrated in Figure 4 below.

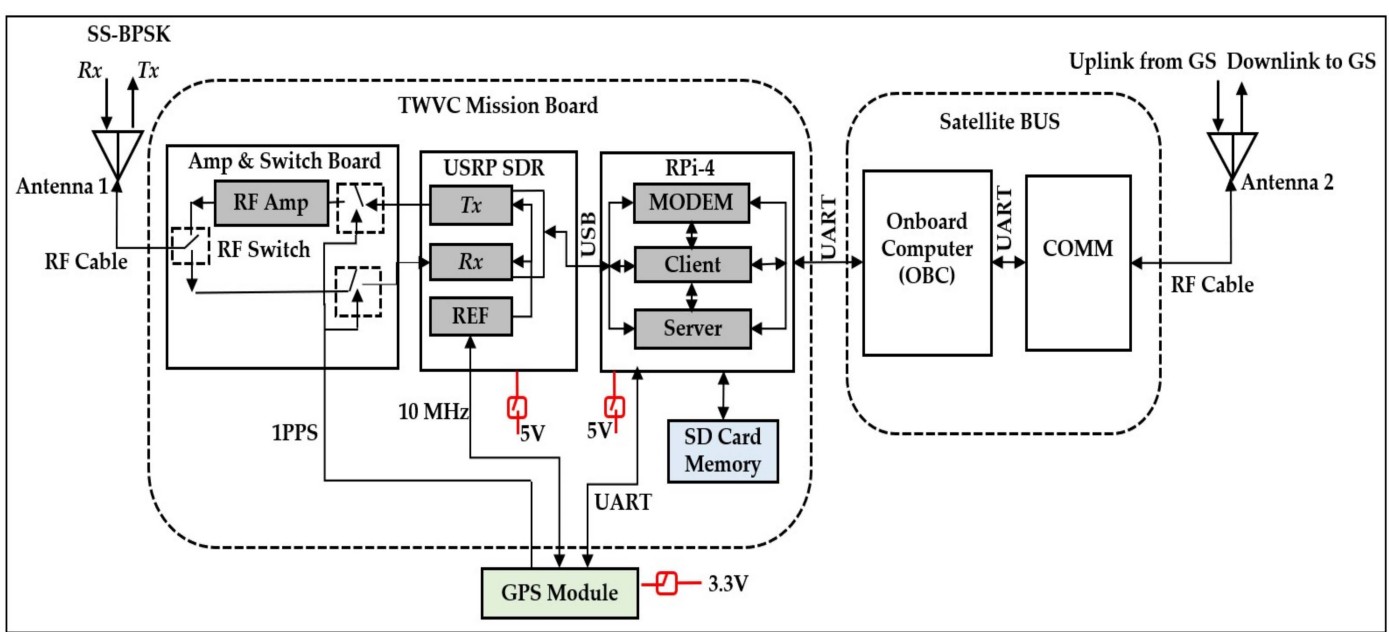

**Figure 4.** System design configuration block diagram.

The satellite bus can be a generic one. The interface between the mission payload and the satellite bus is simply made of power lines (5 V and 3.3 V) and a digital data line universal asynchronous receiver/transmitter (UART). The satellite bus turns on the mission payload based on either the command uplink from the ground station or through reserved commands in the satellite's onboard computer. The mission data will be downlinked to the ground station though the satellite bus. The water vapor mission measurement board is composed of the universal software radio peripheral (USRP) SDR model B205 mini-i, the RPi 4 model B, the amplifier and RF switches board, a GPS module, and the antenna (Antenna 1). Antenna 1 serves as the primary channel of communications. This channel is only used to transmit (*Tx*) and receive (*Rx*) SS ranging signals. Meteorological dual frequency bands ($f_1$ = 400.15 MHz and $f_2$ = 460.00 MHz) with a frequency gap of 59.85 MHz are reconfigured and used for SS transmission and reception. The SDR transceiver is used to transmit and receive the SS signal. To realize mission success in orbit, especially when

the distances between the satellites is large, an RF amplifier is required to amplify the transmitted signal. The RPi 4 conducts digital signal processing on the mission data and is used to control the SDR parameters using GNU Radio software (v3.8.2.0). A GPS module is mounted on each satellite and at the ground stations. The GPS module provides the location, time, a 10 MHz clock, and a one pulse per second (1PPS) signal. The location and time data are carried along with the SS signals. The 10 MHz clock eliminates the SDR's clock and instrument errors at both the transmitting and receiving SDRs. The 1PPS time reference markers are superimposed on to the incoming and outgoing signals through the RF switches. When the GPS modules at both the receiving and the transmitting ends are synchronized, signal propagation time delay in the ISL network can be derived based on the 1PPS time markers. Finally, *TWVC* mission data are deduced from the signal time delay. When the SDR is in reception and transmission modes, the power consumption for this mission subsystem is estimated as 2.81 W and 8.50 W, respectively. Having one SDR to perform both the *Rx* and *Tx* of SS signals at different frequencies, not only saves satellite power but also mass and volume occupied by the mission subsystem compared to systems with multiple radio devices performing the same function.

### 3.1. Procedure to Determine the Time Delay

To detect the time delay, the satellites in the ISL network require SS signals and 1PPS time stamp references for both the outgoing (*Tx*) and incoming (*Rx*) signals. On the receiving side, data transmitted at both frequencies ($f_1$ and $f_2$) is saved as text or recorded wave files. The files are analyzed onboard the satellite to detect the delay following the procedure indicated in Figure 5 below.

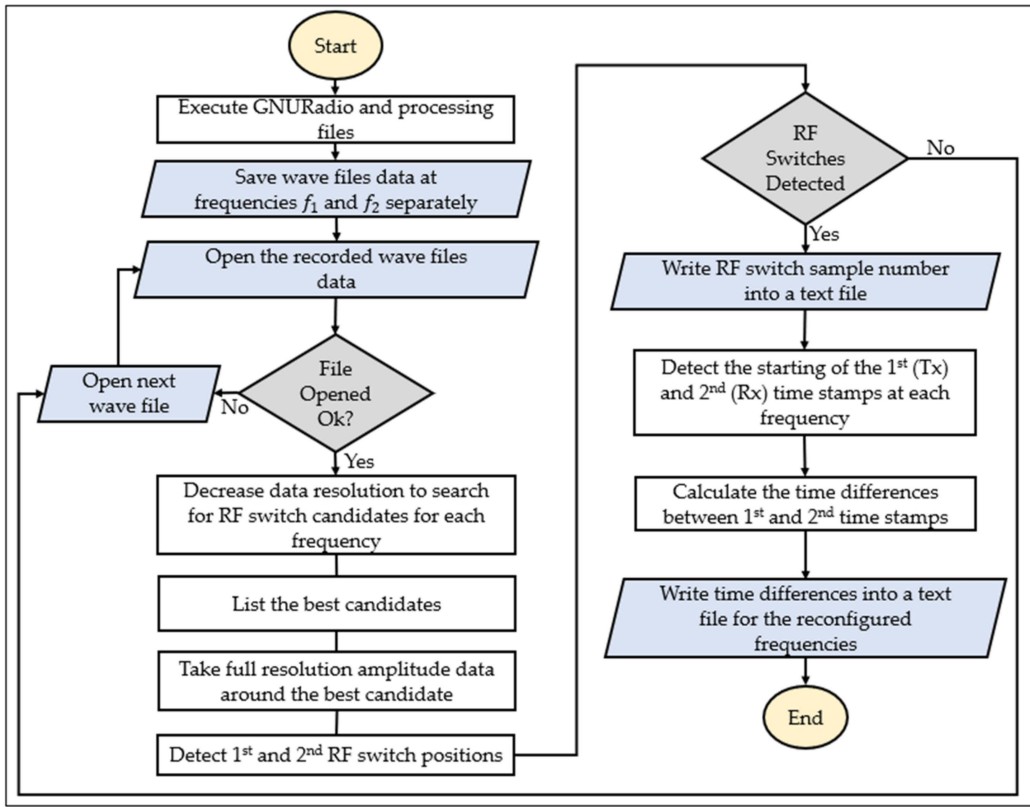

**Figure 5.** Time detection procedure.

### 3.2. Procedure Used to Determine the ISL Frequency Bands

The frequency bands for the water vapor mission are limited to the available meteorological UHF communication frequencies in the international telecommunication union (ITU) region 3 where the Kyushu Institute of Technology, Japan is located [49,50]. To probe

atmospheric water vapor in near-Earth regions using the ISL network, two available options (available band pairs) of frequencies are listed as shown in Table 2.

**Table 2.** Frequency selection table.

| Bands | Band 1 ($f_1$ MHz) | Band 2 ($f_2$ MHz) | Maximum Frequency Gap |
| :---: | :---: | :---: | :---: |
| **Available band pairs** | 400.15–401.00 | 460.00–470.00 | 69.85 |
| **Selected bands** | 400.15 | 460.00 | 59.85 |

From the available options, only one frequency from each of the two pairs can be selected. Considering the mission requirements, lower frequency bands give better estimations of *TWVC* and *TEC*. The reason being that lower bands are influenced much more than higher bands. As a result, two low frequency bands $f_1 = 400.15$ MHz and $f_2 = 460.00$ MHz have been selected from each of the two available pairs. These two frequencies are known and automatically detected by each satellite in the ISL network. Furthermore, the two frequencies have a relatively smaller frequency gap making it feasible to implement with a single antenna that could be tuned to operate at both frequencies while either transmitting or receiving SS signals. Implementation of one antenna to perform both the transmission and reception of the SS signal is proposed to reduce complexity of the design configuration as well as to save on the occupied volume and mass of the satellite. Also, this frequency gap is wide enough to determine and differentiate variation of the *TEC* from the *TWVC* data.

## 4. Functionality Demonstration of the Mission Design Configurations in the ISL Network

Figure 6 shows a simplified test bed configuration used to demonstrate SS transmission, 1PPS time stamping, and frequency reconfiguration between two satellite payloads in an ISL network. Currently, the ISL network operates as a half-duplex system. Also, the satellites operate in two modes for the transmission and reception of SS signals. In Figure 6, Sat *A* is in transmission mode and Sat *B* is in reception mode. In addition to the system design configuration in Figure 4, a delay pulse generator was incorporated at the receiving payload to simulate propagation delay ($\delta T$) due to the *TWVC* and *TEC* in orbit. As a result, the system functionality of deducing atmospheric water vapor and total electron content delay was demonstrated as explained in Section 4.3. Additionally, 30 dB attenuators were used instead of power amplifiers to decrease the transmitted signal power in order to prevent receiving SDR damage or interference from other communication networks. A personal computer (PC) was utilized on one end of the network to serve as the generic satellite bus system for monitoring and acquiring processed data.

### 4.1. SS-BPSK Transmitter

The software for SS-BPSK transmission was implemented on each satellite in the ISL network using GNU Radio digital signal processing software and the Python programming language [51]. The SS signal carries the time and location data of one satellite to another which is needed when computing propagation time delay due to the influence of water vapor and total electron content. Information was transmitted from Sat *A* (SS-BPSK transmitter) to Sat *B* (SS-BPSK receiver) in the ISL network. As illustrated in Figure 7, the SS-BPSK transmitter is where the SS data packets are transmitted from the vector source block over a wider bandwidth at a bit-rate of 250 bits per second, with 1 bit having 250 chips, and 1 chip recurring at every 16 μs. The SS signal is composed of the header, satellite ID, GPS time, and location information as well as the footer. The header and the footer are identifiers which show the beginning and the end of each packet of data conveyed in the communication network. For simulation purposes, the transmitted information was composed of header information: F; satellite identification (Sat ID): 01; GPS data of time (s): 04; latitude (°): 33.53513942; longitude (°): 130.50418427; altitude (m): 46; and footer

information: A. The positive polarity of the latitude and longitude represent north and east positions of the GPS's geodetic coordinates, respectively.

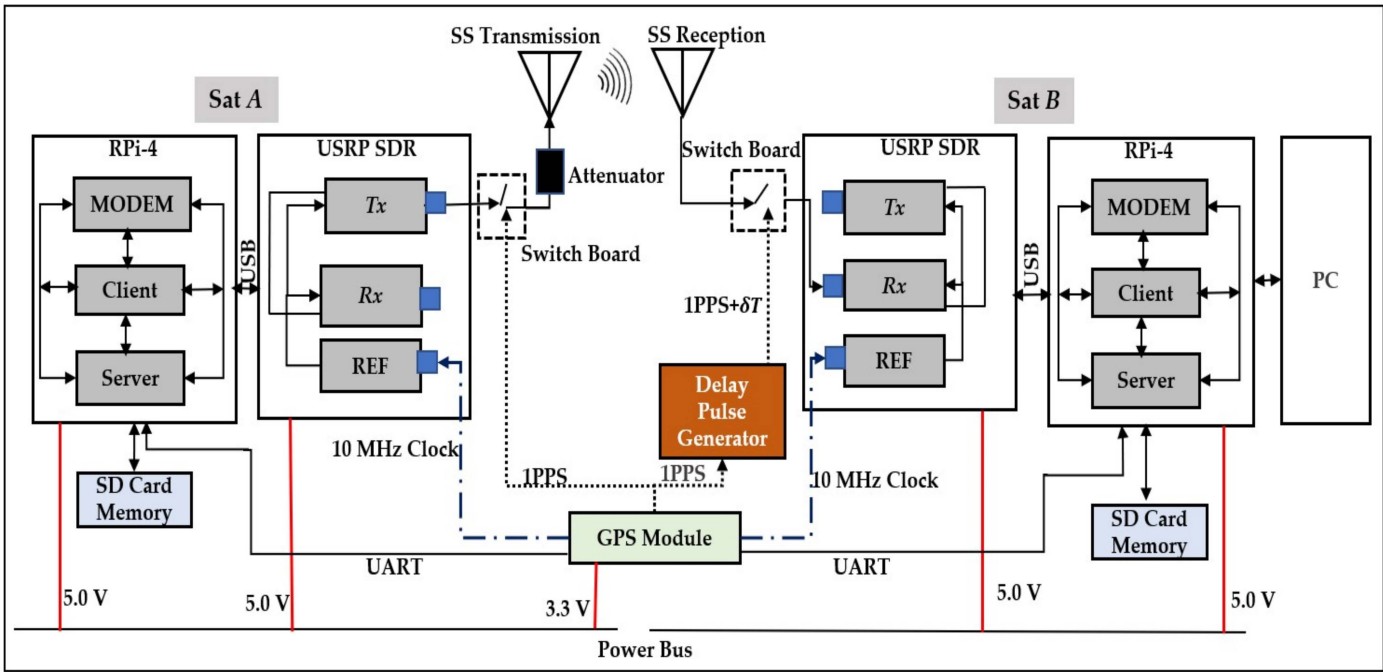

**Figure 6.** ISL network test bed.

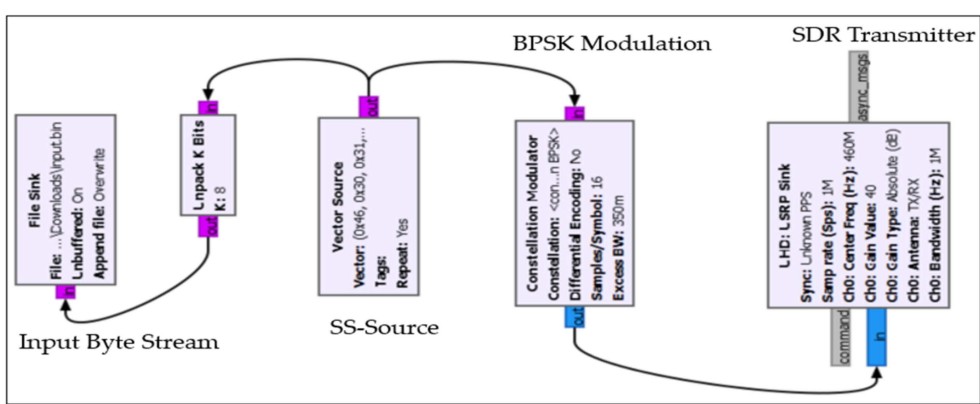

**Figure 7.** SS-BPSK transmitter at Sat *A*.

Likewise, this would indicate the south and west coordinates of the GPS latitude and longitude position data in the case of negative polarity, respectively. The precision the longitude and latitude data were all obtained with ±1.11 mm location accuracy. The data in text was converted to 31 bytes of hexadecimal data (0x46, 0x30, 0x31, 0x30, 0x34, 0x33, 0x33, 0x2E, 0x35, 0x33, 0x35, 0x31, 0x33, 0x39, 0x34, 0x32, 0x31, 0x33, 0x30, 0x2E, 0x35, 0x30, 0x34, 0x31, 0x38, 0x34, 0x32, 0x37, 0x34, 0x36, 0x41) that was transmitted in binary form. The SS source software was implemented to add two more bits (00) to the footer in order to have 250 bits of data transmitted in every second. The original SS data was saved using the file sink block at Sat *A* for comparison with the received data at Sat *B*. The transmitted sequence with GPS data was then modulated with BPSK modulation using a constellation modulator. The modulated data was transmitted with SDR through the USRP Hardware Driver (UHD) sink block. For demonstration purposes sample rates of 1 mega sample per second (MSPS) and 500 kilo samples per second (KSPS) and two frequencies (400.15 MHz and 460.00 MHz) were utilized interchangeably through tuning of the digital signal processing parameters.

### 4.2. SS-BPSK Receiver

Figure 8 shows the implemented GNU Radio-based SS-BPSK receiver software.

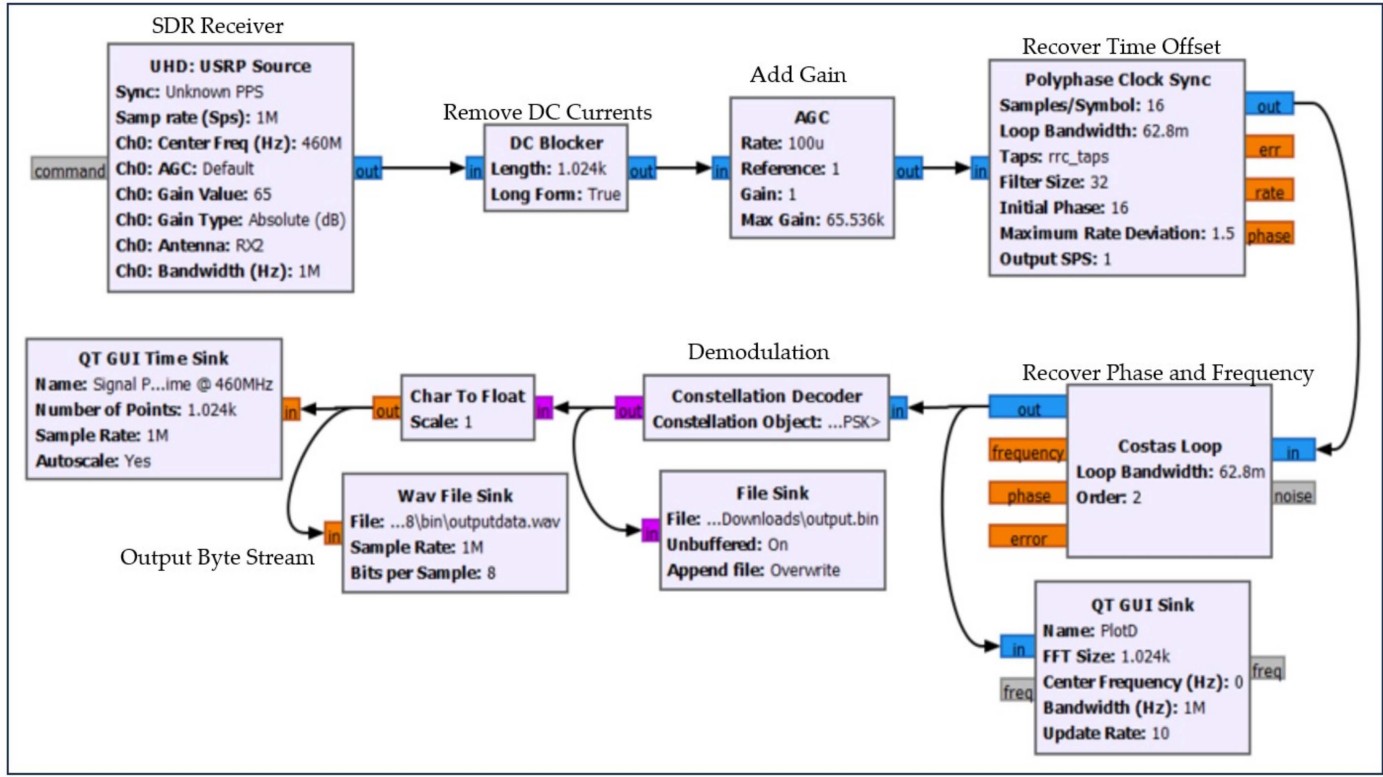

**Figure 8.** SS-BPSK receiver at Sat *B*.

The receiver recovered the signal through the UHD: USRP source block. The DC blocker then eliminated the transceivers' DC offset noise from the signal. After that, the signal went through a high-performance automatic gain control (AGC) unit which regulated the increase in signal amplitude from the original data to the amplified data. Synchronization of the transmitter and receiver time offsets was carried out at the polyphase clock sync block. The Costas loop corrected the phase and frequency offset as well as recovering the carrier. Finally, the signal was demodulated and saved as files in either binary or wave data formats. Figure 9 is the recovered transmitted data in binary form.

**Figure 9.** Recovered SS-transmitted GPS data at Sat *B*.

The saved 250 bits of information received represent header information (1 byte), Sat ID (2 bytes), time (2 bytes), latitude data (11 bytes), longitude (12 bytes), altitude (2 bytes) and footer (1.25 byte). The footer includes 2 bits of information added by the software for efficient transmission. At the receiver the two bits (00) added to the footer by

software for efficient transmission were eliminated. By using the rapid tables binary text translator software [52] to convert the binary data to readable text, the received binary data conformed to the transmitted SS signal data (F, 01, 04, 33.53513942, 130.50418427, 46, *A*). The implemented system could recover the transmitted signal from Sat *A* to Sat *B* and vice versa at both 400.15 MHz and 460.00 MHz in both cases, with and without signal delay. This demonstrated the feasibility of ISL communication where each satellite can transmit or receive location and time data to or from other satellites in the network. In reception mode the satellites could decode and understand the location data of other satellites. As a result, the ISL network requirement was proven.

*4.3. Demonstration of Signal Time Delay Detection and Mission Determination*

The 1PPS GPS reference time stamps from the GPS modules mounted on each satellite were locked to more than 10 satellites and synchronized as shown in Figure 10. On a duty cycle of 1 s, each pulse was detected to be 0.9 s ON and 0.1 s OFF.

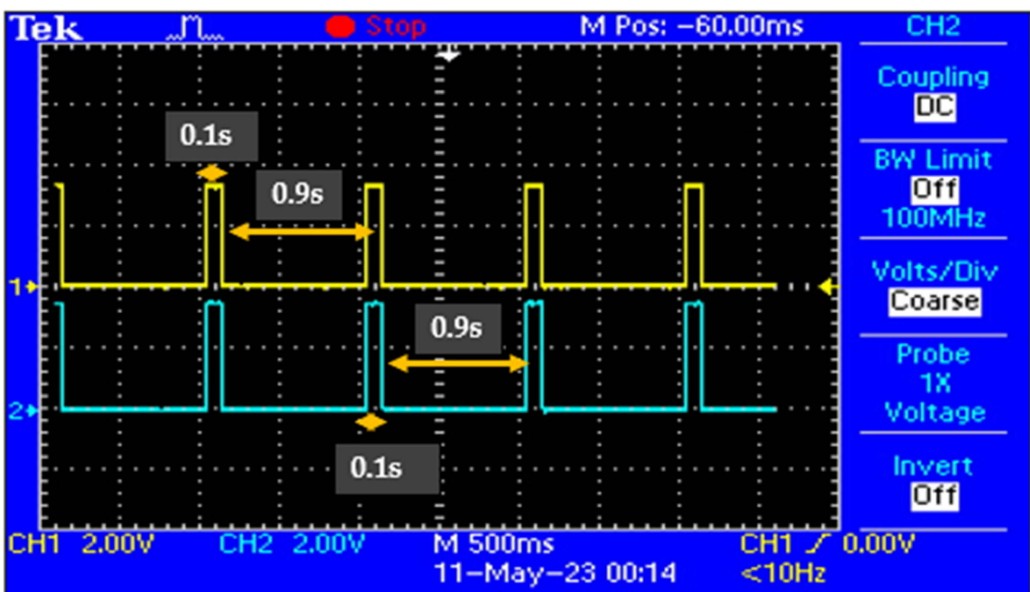

**Figure 10.** Synchronized 1PPS GPS signal for *Tx* (yellow) and for *Rx* (blue).

These 1PPS reference time stamps were marked on both the incoming and the outgoing signals. The saved wave file data shown in Figure 11 was analyzed to visualize and to detect the time stamp and possible signal delay.

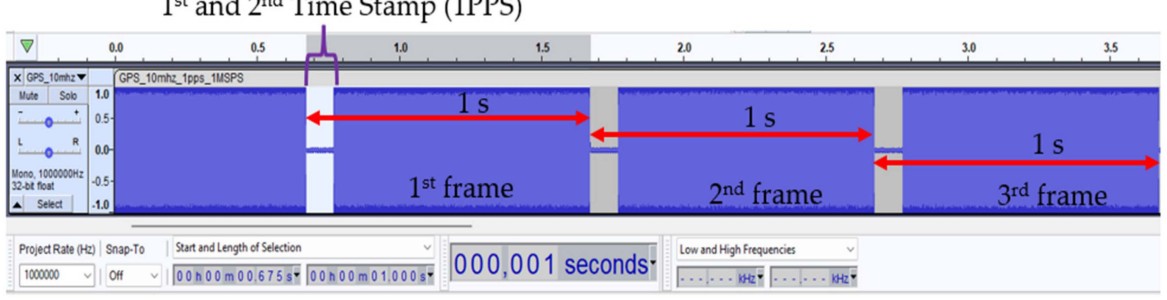

**Figure 11.** Sample of data at 1 MSPS at 460.00 MHz.

The horizontal and vertical axis represent the time and amplitude of the signal, respectively. Three data frames recorded for 3.675 s were observed with a duty cycle of 1 s (0.9 s ON and 0.1 s OFF). Since both satellite payloads were stationary and there was no influence of *TWVC* and *TEC*, the time stamps of the transmitted and the received signal

were superimposed at the same positions. This result signifies that the time stamp function was successful and no delay was detected at both 400.15 MHz and 460.00 MHz frequencies since the signals were free from *TWVC* and *TEC* delays.

During the table satellite test, the SDR hardware suffered from a 35 μs clock jitter error because its internal clock was not fast enough to support the transfer of data. The clock jittering distorted the signals by flipping the bits, inducing noise data or causing time offset by shifting the positions of the bits. To solve the problem, a 10 MHz GPS clock was integrated to the system to eliminate the effects of clock jittering. After the transmission of SS signal from one satellite (Sat *A*), three data frames received on the second satellite (Sat *B*) were analyzed. Figure 12 shows parts of the received data frames when the RF switch was turned ON and OFF. Observations were made on the received data in an interval of every 1 s. The data frames which contain the same information showed that there was no signal distortion due to clock jittering. The positions, patterns, and time for three data frames were analyzed at 0.77330 s, 1.77330 s, and 2.77330 s, respectively. From the analysis, the bit patterns were correctly received without the flipping or shifting of bits, or unwanted noise. Furthermore, the sections of the time stamps when the RF switch is OFF were completely cut off as anticipated. These results demonstrated the efficiency of the system for calculating the *TWVC* data without distortion of information or incorrect predictions due to clock jittering.

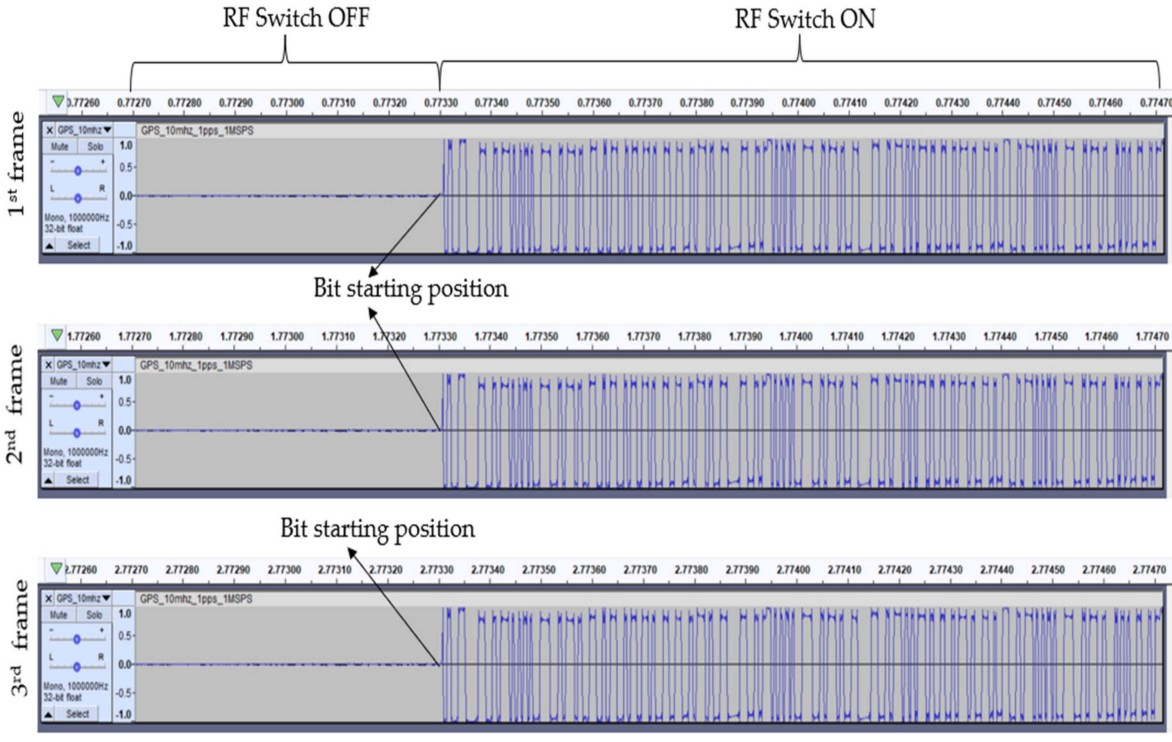

**Figure 12.** Analysis of data frames from the received signal at 450 MHz and 1 MSPS.

Since the position of satellites changes continuously in orbit, the signal passes through a different part of the atmosphere and ionosphere each second. Therefore, a new true range and new propagation time delay should be calculated onboard the satellite. If the signal time shift caused by *TWVC* and *TEC* can be isolated in each second, *TWVC* and *TEC* can be measured from the time delay. In that 1 s, frequency manipulation must be achieved to obtain a high resolution of *TWVC* from *TEC*. To simulate the *TWVC*/*TEC* time delay (s), a delay pulse generator was connected to the receiving side of each satellite as shown in Figure 6. In this study a time delay of $\delta T$ = 0.3 s was assumed in order to clearly visualize the delay simulation functionality and positions of 1PPS time stamps on both the

transmitting and receiving side. This delay was added to a GPS 1PPS signal (yellow) by a delay pulse generator and a delayed pulse (blue) was formed as indicated in Figure 13.

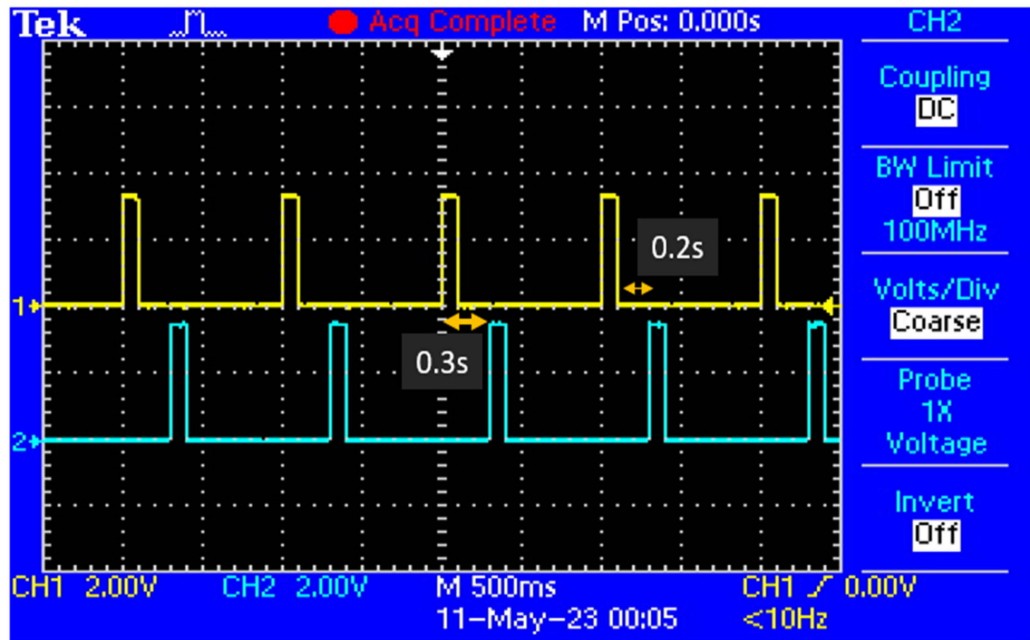

**Figure 13.** GPS 1PPS pulse (yellow) and a delay pulse (blue) output of delay pulse generator.

As observed on the oscilloscope, the time between the rising or falling edges of the 1PPS pulse and the corresponding delayed pulse was 0.3 s as set on the delay pulse generator. Also, the delay between a rising and a falling edge of the two 1PPS signals, d, was observed as 0.2 s.

When Sat *A* transmits SS signals to Sat *B*, the superimposition of these time stamps could be visualized on the received signal as shown in Figure 14.

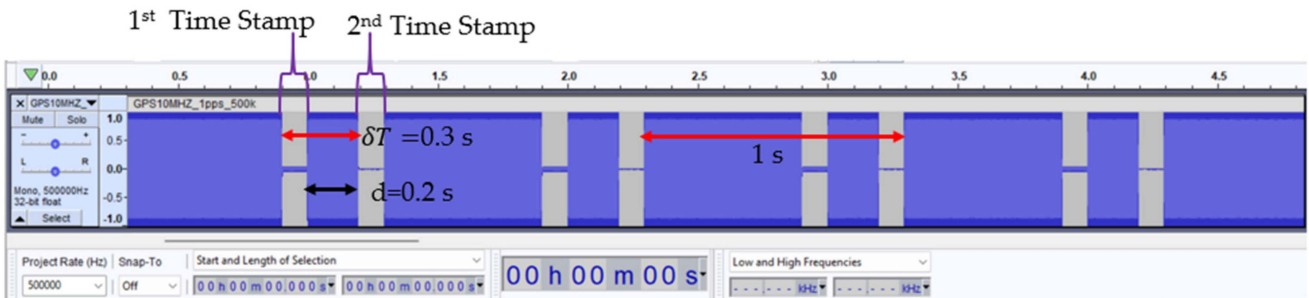

**Figure 14.** Results of the test with delay pulse generator at 400.15 MHz.

Four data frames recorded for 4.85 s were observed. This simulation was performed at 400.15 MHz at 500 KSPS. The superimposed fixed time delays of $\delta T = 0.3$ s and d = 0.2 s, which represent the influence of *TWVC* and *TEC*, were observed on the received data frames. Even though the delays provided by the delay pulse generator were fixed irrespective of frequency change, when the delays were tuned up and down, the variation of the delay change was observed in the data. This proved the efficiency of this system in detecting time delay in orbit where frequency change has an impact on atmospheric geophysical parameters. From the simulation results, the transmitted SS data could also be decoded perfectly even with a delay. For an in-orbit scenario, since *TEC* is inversely proportional to the radio frequencies, the 400.15 MHz frequency is anticipated to give a higher resolution of the measurement values compared to the 460.00 MHz. *TWVC* and *TEC*

difference values are computed with processing algorithms. Finally, *TWVC* can be deduced following Equations (1)–(9).

## 5. Frequency Manipulation and Communication Feasibility

*5.1. Feasibility of Frequency Manipulation*

The communication parameters such as frequency, bandwidth, and sample rate as well as internal SDR transmitter and receiver gains are controlled using the GNU Radio software (v3.8.2.0). To manipulate these parameters during runtime, there is a need to access the inphase and quadrature (IQ) data stream inside GNU Radio Python and/or C ++ blocks. Various frequency reconfiguration techniques are used in satellite communications to optimize performance, increase measurement accuracy, and ensure reliable communication under different conditions. Frequency reconfiguration algorithms used in satellite communications include dynamic spectrum access (DSA) [53], cognitive radio-based frequency reconfiguration [54], frequency-hopping spread spectrum (FHSS) [55], adaptive beamforming and frequency steering [56], and adaptive modulation and coding (AMC) [57]. DSA algorithms allow satellites to adjust their frequency utilization dynamically depending on the availability of spectra in their operational environment. They are nevertheless quite sophisticated in terms of real-time spectrum detection, frequency reconfiguration procedures, and decision making. This complexity augments the likelihood of algorithmic errors. Utilizing cognitive radio approaches, it is possible to identify underutilized or unused frequency bands and opportunistically switch to them. Nonetheless, their spectrum detection, decision making, and reconfiguration impose delays, rendering them unsuitable for real-time applications or crucial communication scenarios.

Moreover, FHSS frequency reconfiguration algorithms provide benefits in terms of security, interference avoidance, and fading resilience, they also present issues in terms of complexity, synchronization, latency, and possible constraints in data rates and bandwidth efficiency. The advantages of adaptive beamforming and frequency-steering frequency reconfiguration algorithms include improved signal quality, reduced interference, and dynamic adaptation; however, maintaining the accurate calibration and alignment of antenna rays are essential for efficient beamforming and steering. Any misalignment significantly degrades performance. AMC frequency reconfiguration methods enhance communication efficiency by dynamically altering the modulation scheme and error-correction coding dependent on the effectiveness of the communication channel. On the other hand, certain AMC algorithms rely on feedback from the receiver to precisely determine the optimal modulation and coding parameters, and this potentially adds communication overhead.

In order to improve performance and reconfiguration speed in real time, ensure reliability, reduce complexity, ensure interoperability in systems and programming languages, and to lower network overhead two algorithms based on server and client architecture were proposed and implemented. These techniques are robust and they achieve the software-based manipulation of SDR and communication parameters in a remote setup of satellites in the ISL network. The first technique is a modified extensible markup language-remote procedure call (modified XMLRPC) and the second one is the TCP/IP method as shown in Figure 15.

In previous versions of the GNU Radio software from 3.7 and below, the XMLRPC algorithm has been implemented to support this function in a server and client setup [44,58]. However, the latest version of GNU Radio from 3.8 and above are incompatible with the traditional XMLRPC client blocks. This study demonstrated an innovation of two techniques that were implemented to grant access to interact with the IQ data stream of GNU Radio Python and C++ modules upon starting, in the middle or at the end of mission execution.

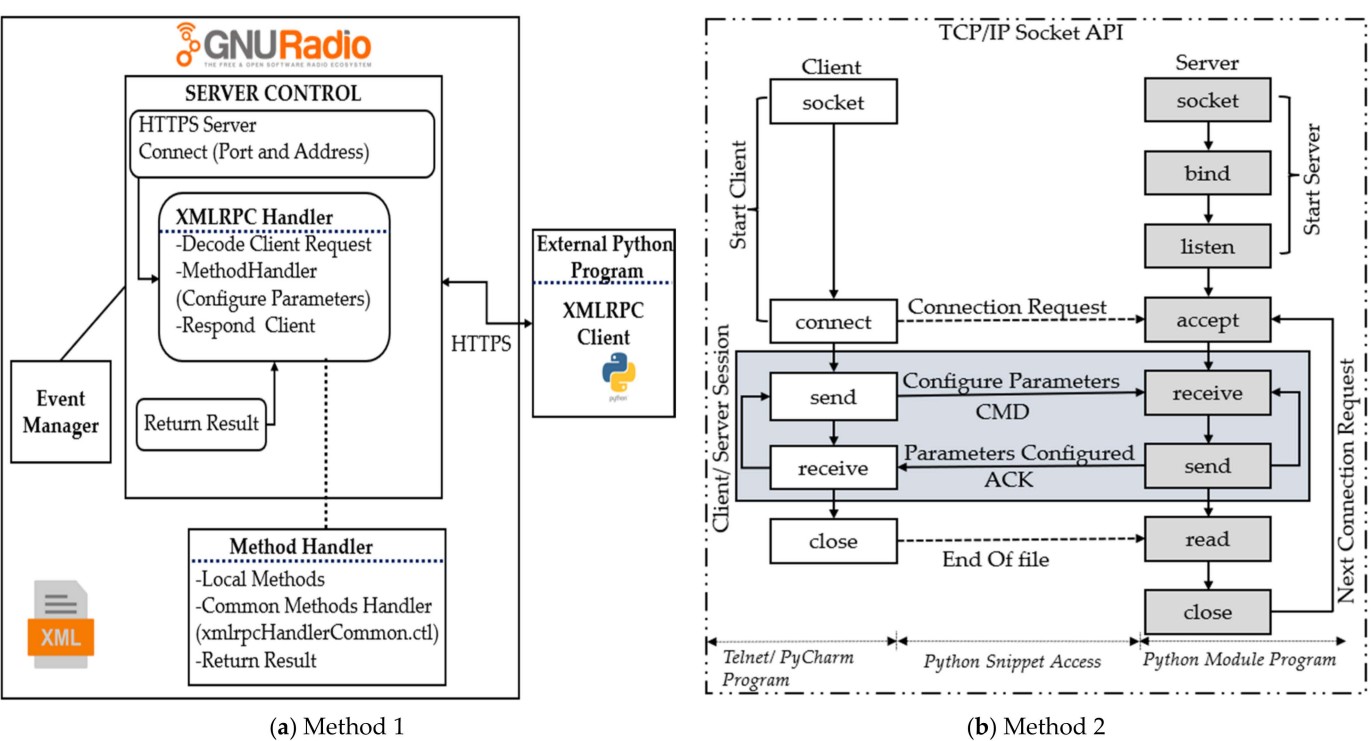

(**a**) Method 1                           (**b**) Method 2

**Figure 15.** Frequency manipulation flow diagrams: (**a**) XML-RPC algorithm architecture; (**b**) TCP/IP algorithm architecture.

### 5.1.1. Procedure of Modified XML-RPC Algorithm

The algorithm makes procedure calls through the hypertext transfer protocol secure (HTTPS) in a server-client relationship as given in Figure 15a. Unlike the traditional client blocks which were typically implemented inside GNU Radio blocks, we modified the concept and implemented an external Python-based client program which is compatible with the latest versions of GNU Radio. The XML-RPC client sends the set messages to the respective XML-RPC server through a reconfiguration command. Acceptance of connection by the server program enables the client's commands to alter SDR parameters inside the GNU Radio. The server facilitates access to an external client-requesting program. This is done through variable call-back functions in the flow graph and procedures that enable tuning of the SDR parameters. The server architecture contains handlers in which the tuning functions and variables are stored. The event manager in the server controls the operation of events or functions requested by the client program. The returned value of the functions is then sent to the client via a secure HTTPS protocol with XML encoding. The encrypted HTTPS uses secure sockets and transport layer security for data integrity and protection.

### 5.1.2. Procedure of the TCP/IP Algorithm

The proposed TCP/IP reconfiguration method also operates in a client-server setup to manipulate SDR parameters. The server program uses a Python module block, and the client program is implemented with an external Python Integrated Development Environment (IDE) such as PyCharm or a Telnet client program [59]. For this program, a Python snippets module was implemented to give the client access to the server program and enable tuning of the SDR GNU Radio parameters. The snippet is executed as a thread that passes self-arguments to access variables from the main GNU Radio flowgraph [59]. Similar to the XMLRPC method, the reconfiguration with TCP/IP method is also feasible at the beginning, during, and at the end of mission execution. The flow graph in Figure 15b describes the TCP/IP architecture and operation procedure.

### 5.1.3. Frequency Manipulation Results and Analysis

Figure 16 demonstrates the frequency translation from the 400.15 MHz to 460.00 MHz peaks and vice versa between Sat *A* and Sat *B*.

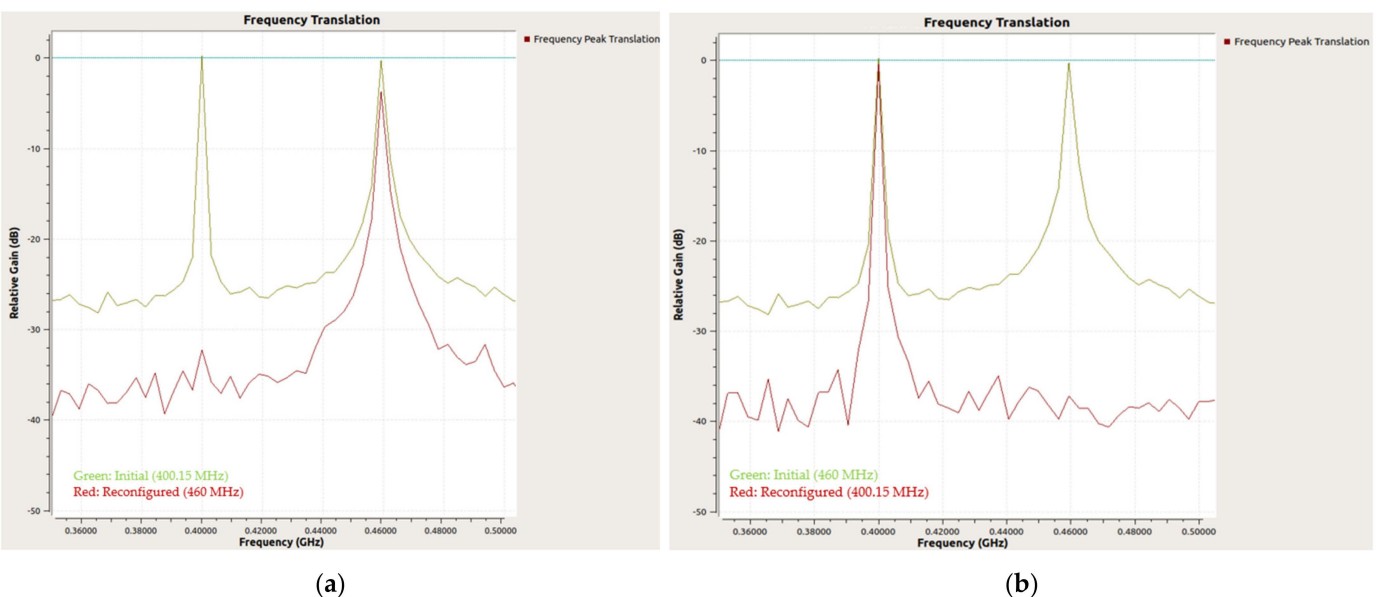

**Figure 16.** Demonstration of frequency manipulation: (**a**) 460.00 MHz to 400.15 MHz; (**b**) 400.15 MHz to 460.00 MHz.

The SDR gains were tuned to attain the transceivers' maximum output power. Initially, the frequency of communication between satellite *A* and *B* was set at 400.15 MHz. A new frequency of 460.00 MHz was set at Sat *A* and SS signals were transmitted to Sat *B*. Sat *B* automatically detected the new frequency and received the SS signal at 460.00 MHz. Observations were made through the spectrums of the transmitted and received signals from both satellites. Also, acknowledgements were received whenever reconfiguration process succeeded. Both satellites in the ISL network showed the same frequency shifts on their spectra. The implemented algorithms can filter SS peaks to visualize the frequency shifts of each peak during tuning and they retain both the previous and newly set frequencies as shown in Figure 16.

Using a combination of the selected SDR as the transceiver and RPi as the processing and control unit, these algorithms achieved frequency manipulation within a time frame of 1 s after tuning. These tests were conducted for 10 s with 1 s accurate frequency transition when mission operations were carried out using 1 MSPS to 3 MSPS sampling rates. However, higher sampling rates >3 MSPS and multiple processing systems running in the desktop version of the Linux operating system (OS) can cause reconfiguration delays and fluctuations in communication. At lower sampling rates, particularly below 1 MSPS, the SDR and RPi can continuously operate by accurately switching the frequencies in 1 s for long periods of operation. Using more efficient microprocessors than the RPi 4 and lighter operating systems that outperform the Linux OS can enhance translation frequency at higher sampling rates. Furthermore, the accuracy of the *TEC* and *TWVC* measurements can significantly improve if the frequency change can be accomplished in less than 1 s.

### 5.2. ISL Communication Feasibility

In order to check the feasibility of the ISL communication network, a link budget has been computed. Several communication parameters of the ISL network including the SDR output power, SDR transceiver sensitivity, antenna gain, and all loses associated with the communication links were measured. Frequencies of 400.15 MHz and 460.00 MHz were considered in this study. The output transmission power of the SDR transceiver and cable

losses were measured as 0.1 W and 0.2 dB, respectively. Sensitivity (dBm) is the minimum power or signal strength that the SDR receiver can detect the transmitted information as having been. The minimum power was obtained by continuously attenuating the transmitted SS signal from Sat *A* with variable attenuators until the receiver at Sat *B* could not decode the SS signals. In order to ensure the integrity and accuracy of the SDR sensitivity, an RF shield box was utilized as shown in Figure 17.

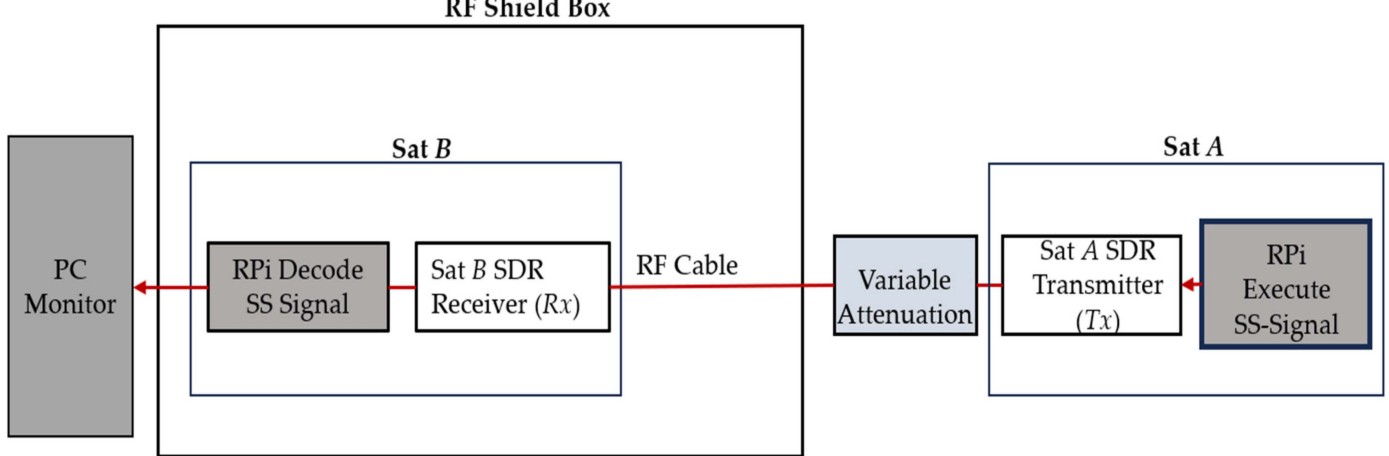

**Figure 17.** Sensitivity test setup.

The RF shield provides a typical measurement environment which eliminates the influence of external unwanted noise, interferences, and internal signal reflections that may distort the signal. Moreover, a cable test environment was utilized to determine the optimum and typical sensitivity of the transceiver. The SDR receiver sensitivity value of −116 dBm was obtained based on the following equation:

$$R_x = T_x - L_{VA} - L_c \tag{10}$$

where $R_x$ (dBm) is the received signal power at Sat *B*, $T_x$ (dBm) is the transmitter output power at Sat *A*, $L_{VA}$ (dB) is the variable attenuation value, and $L_c$ (dB) are RF cable losses. Table 3 summarises the sensitivity test results.

**Table 3.** Sensitivity test results (o: full packets decoded, Δ : partial decoding, x: failed to decode).

| $T_x$ (dBm) | $L_{VA}$ (dB) | $L_{VA}+L_c$ (dB) | $R_x$ (dBm) | Success Rate Based on SDR Internal $R_x$ Gain (dB) | | | | | |
|:---:|:---:|:---:|:---:|:---:|:---:|:---:|:---:|:---:|:---:|
| | | | | 40 | 50 | 60 | 70 | 75 | 76 |
| 20 | −100 | −100.2 | −80.2 | o | | | | | |
| 20 | −120 | −120.2 | −100.2 | x | Δ | o | | | |
| 20 | −130 | −130.2 | −110.2 | | x | Δ | o | | |
| 20 | −135 | −135.2 | −115.2 | | | x | Δ | o | |
| 20 | −136 | −136.2 | −116.2 | | | | Δ | Δ | o |
| 20 | −137 | −137.2 | −117.2 | | | | x | x | Δ |
| 20 | −137 | −138.2 | −118.2 | | | | | | x |

The decoding success rate at Sat *B* was determined based on the SDR internal receiving gain which was tuned from 40 dB to its maximum value of 76 dB. When the variable attenuation was increased and the signal could not be decoded, the SDR gain values were increased until successful decoding occurred. This process was repeated until the SDR

threshold sensitivity of $-116.2$ dBm was reached at the maximum SDR receiver gain of 76 dBi and the decoding processing was no longer possible. The internal SDR transmitting gain at Sat *A* was maintained at its maximum value of 89 dB. A maximum separation distance of 4600 km large enough to probe water vapor profiles near Earth was assumed. Since this distance is large, use of a power amplifier was considered. A power amplifier of 30 W that can be used for small satellite sizes was considered [60]. A dipole antenna with theoretical gain of 2.1 dBi was used in the link budget estimations [61]. Other losses including antenna pointing, polarization, and transmission line losses were measured. The obtained link budget showed that the ISL network has a system link margin of $-9.2$ dB and $-10.4$ dB at 400.15 MHz and 460.00 MHz, respectively. In order to achieve mission success, ways of improving this link budget are explained in detail in the discussion section. Table 4 shows the link budget estimation for this mission.

**Table 4.** Link budget for ISL network.

| Frequency | | MHz | 400.15 | 460.00 |
|---|---|---|---|---|
| **Modulation** | | | **SS-BPSK** | **SS-BPSK** |
| **Data rate** | | **kbps** | **0.25** | **0.25** |
| Satellite *A* (Transmission) | | | | |
| Transmitter Output Power | SDR Output | W | 0.1 | 0.1 |
| | Amplifier Output | W | 30.0 | 30.0 |
| | Total | dBw | 14.8 | 14.8 |
| Gain of Transmitting Antenna | | dBi | 2.1 | 2.1 |
| Transmission Line Loss + Hardware Degradation | | dB | 3.0 | 3.0 |
| Equivalent Isotropic Radiated Power (EIRP) | | dBw | 13.9 | 13.9 |
| Transmission Path | | | | |
| Distance between satellites | | km | 4600 | 4600 |
| Antenna Pointing Loss | | dB | 3.0 | 3.0 |
| Polarization Loss | | dB | 3.0 | 3.0 |
| Atmospheric and Ionospheric Losses | | dB | 1.4 | 1.4 |
| Free Space Loss | | dB | 157.7 | 159.0 |
| Isotropic Signal Level at Spacecraft | | dBw | $-151.3$ | $-152.5$ |
| Satellite *B* (RX Power Sensitivity) | | | | |
| Antenna Pointing Loss | | dB | 3.0 | 3.0 |
| Gain of Receiving Antenna | | dBi | 2.1 | 2.1 |
| Transmission Line Loss + Hardware Degradation | | dB | 3.0 | 3.0 |
| Received Power at LNA input | | dBw | $-155.2$ | $-156.4$ |
| | | dBm | $-125.2$ | $-126.4$ |
| Required Signal power at the Spacecraft | | dBm | $-116.0$ | $-116.0$ |
| System Link Margin | | dB | $-9.2$ | $-10.4$ |

## 6. Discussion

All the mission objectives required to deduce the TVWC as a time delay measurement are demonstrated in this study using low-cost commercial off-the-shelf components. Since this study is a continuation of Kyutech SPATIUM satellites, as mentioned in the objectives of SPATIUM-I and SPATIUM-II [19,29,35], the need for the ISL network to enhance 3D mapping of the atmosphere and ionosphere was demonstrated using this system. SPATIUM-I and SPATIUM-II satellites primarily focused on technology demonstrations for the mea-

surement of *TEC* using a ground-to-satellite communication network. This study deals with both total electron content and atmospheric water vapor measurements using the ISL communication network. Furthermore, the procedure for deducing atmospheric water vapor is presented as a time delay measurement. Kyutech expects to launch a constellation of satellites, which could be more than a thousand satellites, and this makes the desired spatiotemporal resolutions achievable [19]. The ground test results demonstrated that the shortest possible time to reconfigure frequencies and processing has been found to be within 1 s at sampling rates of 1 MSPS and below.

The computed system link margin has a deficiency of at most $-10.4$ dB when the transmission frequency and separation distance between satellites are 460.00 MHz and 4600 km, respectively. It could be improved by implementing state-of-the-art UHF antennas with high gain. Ochoa et al. [62] proposed the use of deployable helical antenna with a gain of approximately 13 dBi, and it could stow into a volume of approximately 0.5U. By adopting a similar antenna, the system link margin becomes positive and ISL communication network could be established between the satellites. Cross polarized linear arrays such as a Yagi-Uda [63–65] also exhibited superior gains of up to 11.5 dBi at UHF frequencies compared to dipole antennas. Using the Yagi-Uda type of antenna could also be considered a solution for this mission. The use of a directive antennas requires attitude control to point the antenna towards the target [66]. In addition, Abulgasem et al. [63] suggested the use of circular polarization in antenna designs for small satellites to eliminate polarization mismatch losses due to antenna misalignment and this guarantees higher link reliability in ISL networks. Furthermore, the *TWVC* mission subsystem consumes approximately 8.50 W in the transmission mode without a power amplifier. Having a power amplifier [60] integrated into the system could significantly improve the system link margin. On the other hand, this would negatively impact the power budget, mass, and volume of the system depending on the limited resources of a small satellite. As a solution, the use of high-power amplifiers demands either the implementation of supplementary power systems such as deployable solar arrays [67], or an increase in the size of the satellite or a reduction of the mission execution time must to meet the requirements of the limitation factors. If these suggested solutions could be implemented, a positive system link margin will be achieved for the proper ISL communications in orbit.

Some limitations due to the processing power of the RPi were encountered as RPi 4 efficiency declined at sampling rates above 3 MSPS; however, the SPATIUM-II satellite achieved the in-orbit demonstration of SS signal demodulation and time delay detection at 1 MSPS based on the RPi CM3+ module, which has a lower processing power compared to the RPi 4 model B [19,35,42]. Also, during ground tests, it was observed that when multiple processes run in the RPi desktop Ubuntu Linux operating system, communication fluctuations and frequency reconfiguration delays occurred. These can be alleviated using high processing power microprocessors, or implementing software interrupt functions which allow the prioritization of important tasks such as frequency reconfiguration and data processing. Using a lighter OS or running the programs in Ubuntu-based servers could also deliver better performance since all system resources would be dedicated only to server tasks rather than having additional desktop resources [68].

The use of precise atomic clocks such as the chip-scale atomic clock (CSAC) or GPS clock [19] to alleviate the clock jittering effect on the SDR or other radio transceivers has been demonstrated in SPATIUM-I and SPATIUM-II. In this study, an external 10 MHZ GPS clock has been used to eliminate the jittering error of 35 μs that was consistently recurring. As a result, all errors associated with the SDR and instruments' clock jittering were removed, and the bits of information could be perfectly received without noise, shifting or flipping of the information bits.

Furthermore, considering a constellation of more than 1000 satellites, large volumes of data will be captured. In such cases, it is recommended to utilize high-data-rate communication networks such as the S-band, C-band, and X-band on the satellite bus for data downlink to the ground [63,69].

## 7. Conclusions and Future Work

The mission measurement concept of *TWVC* is constructed based on radio sensing of the atmosphere. A system design that measures atmospheric water vapor is proposed, the requirements and specifications are determined, and system interfaces and integration are done. A GNU Radio-based transceiver that has the capability of transmitting and receiving SS signal is also implemented. The SS-BPSK has shown good reception and demodulation of SS-transmitted signals with and without delay. All the transmitted data in 1 s intervals are perfectly recovered, and two satellite payloads are utilized to demonstrate the ISL network successfully considering one orbital plane. This research can be further developed to demonstrate the full success criteria of the *TWVC* mission based on a constellation of more than two satellites in motion and in different orbital planes of the LEO. If that is achieved, 3D mapping of atmospheric water vapor or *TEC* can be perfectly deduced. Moreover, this paper only considered the ISL network configuration; however, a combination of fixed and mobile ground stations together with ISL networks can also be implemented to ensure a robust system that determines the 3D mapping of *TWVC* over a wider area. The test results demonstrate the possibility of reconfiguring the frequency onboard the satellite with the proposed algorithms. The determined duration of frequency translation is measured within 1 s as was proposed. The time delay estimation method with the current configuration has also been successful, and it can be recommended as an effective technique for *TWVC* or *TEC* calculations from the signal propagation time delay. For future studies, sweeping through much lower available meteorological frequency bands could be considered as a way to improve mission measurement accuracy. This is because lower frequency bands are more influenced by atmospheric geophysical parameters. However, multiple antennas calibrated for the selected frequencies must be implemented. Also, possible phase delays and errors that may occur in orbit due to the positioning accuracy of the GPS or frequency reconfigurations shall be scrutinized as part of the next objectives. Clock jittering is also successfully removed to ensure the incoming and outgoing signals are not distorted with instrument and clock offsets. This improves mission accuracy within 35 μs when a 10 MHz atomic clock such GPS clock is used. Low orbital attitude, lower frequency bands, and a constellation of more than 1000 satellites can give this method improved accuracy and high spatiotemporal resolutions compared to conventional techniques. Based on these results, the proposed system demonstrated that it is feasible to conduct atmospheric measurements using the radio signals onboard small satellites.

The acquired data will be incorporated into the global atmospheric databases of geophysical parameters for climate and weather prediction models. Participating and contributing to ongoing research initiatives are necessary for integrating obtained data into global atmospheric databases for climate and weather prediction models, in addition to adherence to the accepted standards and data sharing agreements. The 3D-mapped data of the *TWVC* will be aligned with standardized formats, units, and metadata acknowledged by the atmospheric research community. Additionally, partnership with organizations that specialize in collecting and distributing atmospheric data will be considered. To guarantee seamless integration of the acquired data with existing global atmospheric databases, data quality control will be conducted by institutions that specialize in data evaluation and validation. Institutions and researchers involved in atmospheric modelling and research will be granted open access to the acquired data. Finally, the institute will hold workshops and forums that bring together researchers, modelers, and data providers to discuss integration challenges, strategies, and opportunities. This research will be implemented in future Kyutech satellite projects involving radio sensing of the atmosphere.

**Supplementary Materials:** The following supporting information can be downloaded at: https://www.mdpi.com/article/10.3390/aerospace10090807/s1.

**Author Contributions:** Conceptualization, R.M.N.; methodology, R.M.N.; hardware and software, R.M.N.; validation, R.M.N., N.C.O., M.C., and D.N; formal analysis, R.M.N., N.C.O. and M.C.; investigation, R.M.N.; resources, M.C.; data curation, R.M.N.; writing—original draft preparation,

R.M.N.; writing—review and editing, N.C.O., D.N. and M.C.; visualization, R.M.N.; supervision, M.C.; project administration, M.C.; funding acquisition, M.C. All authors have read and agreed to the published version of the manuscript.

**Funding:** This research received no external funding.

**Informed Consent Statement:** Not applicable.

**Data Availability Statement:** Data is contained within the article and Supplementary Materials.

**Conflicts of Interest:** The authors declare no conflict of interest.

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
