# Peer review of "A Study on the Derivation of Atmospheric Water Vapor Based on Dual Frequency Radio Signals and Intersatellite Communication Networks"

_aerospace, doi:10.3390/aerospace10090807_

Round 1

Reviewer 1 Report

The article titled "A Study on Derivation of Atmospheric Water Vapor Based on Dual Frequency Radio Signals and Intersatellite Communication Networks" presents an interesting approach to deriving atmospheric water vapor content using spread spectrum (SS) radio signals and software-defined radio (SDR) technology on low Earth orbiting (LEO) satellites. The study proposes the use of a constellation of small satellites with intersatellite communication (ISL) networks to achieve three-dimensional (3D) mapping of water vapor content. While the proposed technique is innovative and addresses some important challenges, there are several critical comments to consider:

·         How does the accuracy of the proposed method compare to existing techniques like GNSS in measuring atmospheric water vapor content?

·         Could you elaborate on the specific challenges and potential solutions associated with implementing a constellation of small satellites with ISL networks? How do you plan to address issues related to satellite deployment, communication, and long-term maintenance?

·         In the validation process, how closely do the simulation and ground test results align with actual in-orbit conditions? Are there any discrepancies, and how do these affect the method's reliability and accuracy?

·         Could you expand on the advantages and limitations of using low-cost SDR technology for the proposed method? How might the use of higher-powered microprocessors or different antenna configurations impact the results?

·         You briefly mentioned the potential applications of the proposed method in telecommunications, broadcasting, and internet networks. Can you provide more insights into how the findings of this study could be leveraged in these domains?

·         Considering the complex nature of atmospheric conditions and the potential for interference, how do you account for factors such as signal distortion, noise, and other external influences in the measurements?

·         Could you provide more details on the specific algorithms or techniques used to distinguish between total water vapor content (TWVC) and ionospheric total electron content (TEC)? How robust are these algorithms in real-world scenarios?

·         The article discusses a constellation of more than 1,000 satellites. How do you plan to manage such a large number of satellites effectively, and what challenges might arise from coordinating their movements and data communication?

·         In terms of practical implementation, how would you ensure that the proposed system is compatible with existing satellite technologies and infrastructures, both in terms of hardware and software?

·         Could you provide more information on the comparison between the proposed method and traditional GNSS techniques in terms of accuracy, temporal resolution, and spatial coverage? How do these trade-offs impact the overall suitability of the method for climate and weather prediction models?

·         Can you elaborate on the potential impact of clock jittering on the proposed method's accuracy and the steps taken to mitigate its effects? How sensitive is the method to small timing errors, and how might these errors affect the final measurements?

·         Considering the future scalability of the proposed technique, what strategies do you have in mind for maintaining the accuracy and reliability of measurements as the number of satellites in the constellation increases?

·         How do you plan to integrate the acquired data into global atmospheric databases for climate and weather prediction models, and what kind of collaboration or coordination with existing research initiatives will be required?

·         Given the potential for disruption or interference in communication networks, what backup or redundancy measures have been considered to ensure data collection and transmission remain uninterrupted?

Author Response

Dear Reviewer 1

Reviewer 2 Report

The article proposes an original method for determining the global atmospheric total water vapor content (TWVC) based on the use of a network of small satellites on which P-band transceivers are installed. Signal phase delay measurements performed at two frequencies allow to separate the contribution of tropospheric water vapor and ionospheric electron density, and subsequently retrieve TWVC value. Methodical issues of radio signal measurement, frequency manipulation and digital processing are discussed in detail. The authors consider a constellation of more than 1000 small low-orbiting satellites and a software-defined radio (SDR) technology developed to achieve global 3D mapping of troposphere and ionosphere with improved spatial resolution.  At the same time, there is a serious remark under the article. There is no analysis on phase delay measurement errors and further TWVC retrieval accuracy presented. Errors in phase delay evaluation may arise due to inaccuracies in determining the location of satellites, neglecting the influence of spatio-temporal inhomogeneities of the electron density on the signal propagation path between the satellites. Additional errors can occur due to peculiarities of the dual-frequency method for solving the inverse problem of TWVC retrieval. The paper requires a major revision before publication.

Author Response

Dear Reviewer 2

Reviewer 3 Report

Review

Overall, the paper titled "A Study on Derivation of Atmospheric Water Vapor Based on Dual Frequency Radio Signals and Intersatellite Communication Networks" presents an intriguing approach to predicting and measuring atmospheric total water vapor content (TWVC) using spread spectrum (SS) radio signals and software-defined radio (SDR) technology on low Earth orbiting (LEO) satellites. The proposed method introduces the utilization of intersatellite communication networks for three-dimensional (3D) mapping of water vapor content, taking into account the potential interference of ionospheric total electron content (TEC). The paper is well-structured and presents a comprehensive analysis of existing techniques before delving into the details of the proposed technique.

Here are some specific comments and suggestions for improvement:

Abstract: The abstract provides a concise overview of the paper's contents. However, it could benefit from including a brief statement about the novelty of the proposed method and its potential contributions to the field of atmospheric remote sensing and numerical results.

Introduction: The introduction effectively highlights the significance of atmospheric water vapor content and its impacts on climate change, weather patterns, and various phenomena. It would be helpful to clearly state the gap or limitation in current measurement techniques that the proposed method aims to address. However, it is necessary to refer to some previous works regarding the relation between water vapor, tomography, precipitation and temperature and climate change add them to the literature:

  • A link between surface air temperature and extreme precipitation over Russia from station and reanalysis data
  • Impact of Climate Change Parameters on Groundwater Level: Implications for Two Subsidence Regions in Iran Using Geodetic Observations and Artificial Neural Networks (ANN)
  • Function-Based Troposphere Tomography Technique for Optimal Downscaling of Precipitation
  • Evaluating the Dependence between Temperature and Precipitation to Better Estimate the Risks of Concurrent Extreme Weather Events

Proposed Technique Analysis: The section describing the proposed technique is well-detailed and logically structured. However, it might be beneficial to provide a concise summary of the key steps of the proposed technique after the detailed description. This would help readers quickly grasp the overall methodology.

Quantitative Results: The paper could benefit from including more quantitative results to support the system's performance claims. Providing metrics such as bit error rate (BER), signal-to-noise ratio (SNR), and signal distortion levels would strengthen the experimental validation.

Comparison with Existing Methods: It might be helpful to provide a brief comparison of the proposed frequency manipulation methods (modified XML-RPC and TCP/IP) with existing methods, discussing advantages, drawbacks, and potential scenarios where each approach is most suitable.

Discussion: In the final part of the discussion, reiterate the key findings and contributions of the study, emphasizing how the proposed system aligns with Kyutech's broader satellite projects. Highlight how the results can contribute to atmospheric data collection and its integration into global databases.

Author Response

Dear Reviewer 3

Round 2

Reviewer 1 Report

The authors have addressed the comments. Paper may accepted.

Reviewer 2 Report

Although the analysis of TWV retrieval accuracy has not been completely fulfilled, the manuscript has been significantly improved. I believe that the manuscript, in its new version, should be recommended for publication in Remote Sensing journal.  

Reviewer 3 Report

The paper could be published in the present form.